# Effectiveness of Evaluation of Adenoid Hypertrophy in Children by Flexible Nasopharyngoscopy Examination (FNE), Proposed Schema of Frequency of Examination: Cohort Study

**DOI:** 10.3390/diagnostics12071734

**Published:** 2022-07-17

**Authors:** Aleksander Zwierz, Krzysztof Domagalski, Krystyna Masna, Paweł Burduk

**Affiliations:** 1Department of Otolaryngology, Phoniatrics and Audiology, Faculty of Health Sciences, Ludwik Rydygier Collegium Medicum, Nicolaus Copernicus University, 85-168 Bydgoszcz, Poland; krymasna@gmail.com (K.M.); pburduk@wp.pl (P.B.); 2Department of Immunology, Faculty of Biological and Veterinary Sciences, Nicolaus Copernicus University, 87-100 Torun, Poland; krydom@umk.pl

**Keywords:** adenoid size, adenoid hypertrophy, adenoidectomy indications, flexible nasofiberoscopy

## Abstract

Objectives: To demonstrate the effectiveness of endoscopic assessment of the pharyngeal tonsil in defining the size of the adenoid hypertrophy in endoscopic examination that would be equivalent to intraoperative assessment as a large adenoid, and to determine the lowest necessary frequency of tests to assess the variability of its size. Methods: The study is based on an analysis of two groups of children diagnosed and treated in a children’s ENT outpatient clinic and ENT department. In the first group, adenoid size was assessed based on flexible endoscopy, and then with a consequent adenoid surgery with assessment of the intraoperative size of the adenoid, we compared the size of the adenoid. The second part of the study included a group of 81 children. We analyzed performed flexible nasopharyngoscopy examinations (FNE) of each child, and compared the change of adenoid size in a minimum of two subsequent examinations over a period of 1 year or more. Results: The sensitivity of flexible endoscopic examination in the assessment of the pharyngeal tonsil was determined at 97.3%, and specificity at 72.7%. The ROC curve shows the value of adenoid-to-choana (A/C) ratio as 75% or more in the preoperative FNE, indicating that the tonsil during surgery is assessed as large. Among the children, 26.3% had a change in adenoid size of more than 15% in the A/C ratio in 1 year of observation, and 45% of the children had A/C ratio changed above 15% in a period of 3 years of observation. Conclusions: FNE examination is highly effective in assessing the size of the pharyngeal tonsil. We proposed a schema for frequencies of FNE examinations and treatment dependent on A/C ratio and worsening of ailments.

## 1. Introduction

The adenoid (pharyngeal tonsil) has been a mysterious cause of numerous ailments in children for years. It is perceived as the cause of the recurrence of upper respiratory infection, snoring, hypoacusis and otitis media with effusion, obstructive sleep apnea, facial growth disorders, malocclusion, and, finally, may have an influence on behavioral symptoms [1,2,3]. Problems of examination and visualization of the adenoid make it difficult to assess its true role in these disorders. Hans Wilhelm Meyer was the first to diagnose adenoid hypertrophy with his finger in 1868, and then removed the adenoid [4]. Since then, numerous diagnostic techniques have been introduced, seeking the most effective, but also the most comfortable and least burdensome for the patient. These techniques may be divided into two groups: invasive and imaging. The first group consists of manual finger or mirror examination through the mouth, rigid or flexible nasopharyngoscopy examination (FNE), videofluoroscopy, and acoustic rhinomanometry. The second group comprises ultrasonography, lateral nasopharyngeal X-ray (lateral cephalogram), multirow detector CT with virtual laryngoscopy, or MRI of the nasopharynx [5,6,7,8,9]. However, the effectiveness of these tests is related to unobjective results, such as the severity of adenoid hypertrophy symptoms, otitis media with effusion (OME), tympanometry, or air flow through the nose. However, not all adenoid hypertrophy symptoms must be associated with adenoid hypertrophy. In some cases, it may be simulated by other causes of nose and nasopharyngeal obstruction, such as septal deviation, nasal polyps, nasal concha hypertrophy, and allergic rhinitis [6,10]. Some of the adenoid examination methods refer to intraoperative evaluation using a transoral mirror examination or to nasal endoscopy [6,9,10,11]. Patel et al. showed that in patients who had less obstructive adenoid hypertrophy with less than 75% in adenoid-to-choana scale (A/C ratio), preoperative FNE and intraoperative mirror exam may not correlate, suggesting that intraoperative mirror examination performed in a horizontal position in anesthesia with relaxation may also be fraught with observation errors [6]. We should ask the following question: How accurate is the FNE examination as a defined gold standard? Therefore, in the following work, an attempt was made to relate the results of the endoscopic examination to the size of the tonsil being removed during the surgery. Another issue is the question of how often in the clinical observation should such an examination be performed, and more precisely, the question should be answered for how the size of the tonsil changes with age. There is prevalent opinion, often based on ENT doctors’ experience or based on the almost 100-year-old Scammon’s theory in accordance with proposed Scammon curves, that the adenoid undergoes hypertrophy during childhood and involution in adulthood [12]. To the best of our knowledge, only three longitudinal observational studies, one performed in 1976 by Handelman and two performed in Japan, published in 2018 and 2021, have assessed the sizes of the adenoid, all based on lateral cephalometric radiography [13,14,15]. Therefore, there is the need for longitudinal adenoid observations to investigate the process of adenoid involution, and to assess its influence on the reduction of adenoid symptoms in children. We used FNE to analyze the term change of the adenoid size.

The aim of this study was to determine the effectiveness of endoscopic assessment of the pharyngeal tonsil and its correlation with intraoperative assessment of adenoid size. The second aim of this study was to analyze longitudinal changes in adenoid size to obtain knowledge of its yearly variations. This may be useful in determining the lowest necessary frequency of tests to monitor the variability in its size.

## 2. Materials and Methods

### 2.1. Research Participants

The study is based on a retrospective analysis of two groups of children diagnosed and treated in a children’s ENT outpatient clinic. The first group consisted of 108 children aged 3 to 9 who were qualified and underwent subsequent adenoidectomy because of adenoid hypertrophy and recurrent adenoid symptoms, obstructive sleep apnea (OSA), and as an adjuvant treatment of otitis media with effusion with grommet insertion between 2019 and 2021. The second analyzed group consisted of children who were the patients of the ENT outpatient clinic for a longer period of time, at least 1 year, and had undergone subsequent endoscopic choana examination between 2016 and 2021.

### 2.2. Inclusion and Exclusion Criteria

In the first part of the study, we included children admitted to the ENT outpatient clinic who subsequently underwent adenoidectomy or adenoidectomy with grommet insertion. In the second sample, we enrolled patients who did not undergo adenoid surgery during the observation period. We excluded patients with genetic diseases (Down, Treacher–Collins Syndrome, 9th chromosome trisomy) and craniofacial anomalies, cleft palate, or submucosal cleft from the studies. Children who had an active upper respiratory infection during the performed endoscopy or those who had previously undergone adenoidectomy or maxillofacial surgery or trauma were eliminated from the study.

### 2.3. Study Methods

In the first analyzed group, adenoid size was assessed based on flexible endoscopy and consequent adenoid surgery with use of an age-appropriate adenotome. We analyzed 108 subsequent adenoidectomies. Before each surgery, ENT doctors (PB, KM) analyzed the endoscopy previously performed in the ENT outpatient clinic and assessed the adenoid size in adenoid-to-choana ratio (A/C) as a percentage. All adenoidectomies were performed by a third ENT doctor (AZ), who was blind to the results of the endoscopies. Intraoperatively, the surgeon assessed the adenoid size as “small,” “medium,” or “large.” The assessment of the size of the tonsil during surgery was expressed as “large” if the removed tonsil tissue was larger than the hole in the Beckmann adenotome. “Medium” was smaller than the size of the adenotome opening, but occupying more than three-fourths of its lumen. “Small” tonsil tissue occupied less than three-fourths of the adenotome opening (Figure 1). The intraoperative size of the tonsil was compared with an endoscopic assessment. We also performed control visits at least 2 weeks, 6 months, and 12 months after surgery, asking parents if the children had any problems with breathing through the nose, presence of snoring, mouth-breathing, and hyponasal voice or recurrent middle-ear disease; and in any suspicion of adenoid regrowth, we performed flexible endoscopy. If we observed that regrowth of the adenoid tissue to the A/C ratio reached 30% without middle-ear disturbance symptoms, we did not perform revision surgery.

The second part of the study included a group of 81 children. We analyzed endoscopic examinations (FNE) performed by one children’s ENT doctor (AZ) on each child and compared the change in adenoid size in a minimum of two subsequent examinations over a period of 1 year or more.

### 2.4. Endoscopy

Flexible nasopharynx endoscopic (FNE) examinations were performed in the ENT outpatient clinic by an ENT specialist using the Karl Storz Germany Tele Pack compact endoscopy system (18 kilo pixels, 2.8 mm outer diameter, flexible nasopharyngoscope; Medit Inc.). Based on the recorded video files, we used DaVinci Resolve 17 software (Blackmagic Design) to evaluate and calculate the percentages of obturation of the choanae (A/C ratio, adenoid-to-choana ratio as a percentage). The A/C ratio was assessed with an accuracy of up to 5%. Additionally, we classified endoscopic adenoid size in accordance with the section of the Bolesławska scale: grade I, adenoid tissue filling less than one-third of the vertical portion of the choanae; grade II, adenoid tissue filling between one-third and two-thirds of the choanae; and grade III, adenoid tissue filling more than two-thirds of the choanae [16].

### 2.5. Surgery

Surgery was performed under general anesthesia. The child’s mouth was opened widely with a McIvor retractor. Next, the palate was palpated for evidence of submucous cleft. Then, the catheter was inserted into the nose, retrieved through the mouth, and pulled anteriorly to retract the soft palate forward. The adenoid was removed through the mouth with the use of a Beckmann adenotome. The size of the adenotome used was selected according to the age of the child, in accordance with Shaalan’s recommendations (3–4 years: 18 mm; 5–6 years: 19–20 mm; 7–9 years: 20–21 mm) [17]. A cotton pledget was used to stop any bleeding. Indirect mirror visualization of the choana was performed at the end of the surgery to confirm complete adenoid removal.

### 2.6. Statistical Analysis

Quantitative data are presented as mean ± standard deviation (*SD*) and median with interquartile range (Q25–Q75). For the categorical variables, we used numbers (*n*) and percentages (%). Differences in the distribution of categorical variables were evaluated using Pearson’s χ2 or Fisher’s exact test, as appropriate. Student’s *t* test was used for comparison of quantitative variables between two independent groups. For comparing more independent groups, a one-way analysis of variance (ANOVA) was used. The difference between the maximum and minimum values (range) of the A/C ratio in each patient was used to evaluate changes in the adenoid size over time. In order to determine the diagnostic value of the A/C ratio to detect the operative adenoid size, the receiver operating characteristic (ROC) method was used, analyzing the area under the curve (AUC), as well as the sensitivity, specificity, positive predictive value (PPV), and negative predictive value (NPV) for the selected cutoff.

For all these tests, two-tailed *p* values were used, and a *p* value < 0.05 was considered statistically significant. The Bonferroni correction for multiple testing was used. All statistical analyses were performed with SPSS software (Statistical Package for the Social Sciences version 26, Armonk, NY, USA).

### 2.7. Ethics

Ethical approval for this study was obtained from the ethics committee of Nicolaus Copernicus University (KB 136/2022).

## 3. Results

### 3.1. Association of A/C Ratio with Operative Adenoid Size

#### 3.1.1. Patients’ Characteristics

We performed subsequent adenoidectomies in 108 children (43 girls and 65 boys, mean age 5.2 ± 1.8 years) who were previously endoscopically evaluated for A/C ratio and Bolesławska scale [16] (Table 1). The range of A/C ratio was 40% to 95%; the majority of children had 80%, 85%, and 90% A/C ratio (21.3%, 20.4%, and 14.8%, respectively). In our study, 23 (21.3%) children were classified as grade II adenoid hypertrophy by Bolesławska scale, and 85 (78.7%) were classified as grade III. During the surgery, 75 (69.4%) of the removed adenoids were assessed as large, 12 (11.1%) as medium, and 21 (19.4%) as small (operative adenoid size). There were no statistically significant differences between gender and age. The mean age of the girls was 5.4 years, and that of the boys was 5 years. In our sample, we did not report gender-dependent differences in the prevalence of adenoid hypertrophy in Bolesławska scale and adenoid size reported during the surgery. In our study, eight (18.6%) girls and 15 (23.1%) boys were classified as grade II by Bolesławska, and 35 (81.4%) girls and 50 (76.9%) boys were classified as grade III. During the surgery, the adenoids of eight (18.6%) girls and 13 (20.0%) boys were classified as small, six (14.0%) in each gender were classified as medium, and 29 (67.4%) in girls and 46 (70.8%) in boys as a large operative adenoid size. Detailed data for the whole group and according to gender are presented in Table 1.

#### 3.1.2. Diagnostic Value of A/C Ratio

Comparing the size of the tonsil assessed on the A/C ratio and Bolesławska scales with the macroscopic assessment of the removed adenoid depending on whether it was assessed as large or medium and small, a high agreement was shown (*p* < 0.001 for all) (Table 2).

Based on the ROC curve, the diagnostic value of A/C ratio in large operative adenoid size diagnosis was evaluated (Figure 2). In our study group, high diagnostic accuracy was found for A/C ratio (AUC = 0.894; 95% CI = 0.818–0.969; *p* < 0.001). The course of the ROC curve indicates the existence of two potential cutoff points for the A/C ratio with significant discrimination values: 75% (<75% vs. ≥75%) and 80% (<80% vs. ≥80%). According to the determined cutoff, a higher A/C ratio in the endoscopic examination indicates that the adenoid size during surgery is assessed as large. For the cutoff value of 75%, the sensitivity was 97.3%, and the specificity was 72.7%. For A/C ratio at the cutoff value of 80%, the sensitivity was 84.0% and the specificity was 81.8% (Table 3). Among these two threshold values, the A/C ratio of 75% has greater prognostic accuracy in detecting a large operative adenoid size.

### 3.2. Volatility of the A/C Ratio over Time

The second retrospective study group of patients consisted of 81 children; 43 (53.1%) were girls and 38 (46.9%) were boys. All patients’ characteristics data are presented in Table 4. The mean age of the children was 3.9 years (*SD* ± 1.2). Forty-seven (58.0%) children were examined three times, and others more often. The mean period of observation was 2.4 years. In this analyzed group, 38 (46.9%) children were examined three times over a period of 1 year. Fifty-seven (70.4%) children proceeded with flexible nasopharyngoscopy at least one time per year. Thirty-seven (45.7%) children were subsequently examined in a 2-year period, and 20 (24.7%) in a 3-year period.

Only 7.9% of the children had a change in tonsil size of more than 15% on the A/C scale in 1 year of observation (Table 4). In the group of 37 children examined at least three times over a period of 2 years, eight (21.6%) of them had a change in the size of the tonsil above a 15% A/C ratio. In both groups of patients, less than half of the adenoids were changed more than 10% in A/C ratio scale (26.3% and 43.2%, respectively). From the group of 20 children with at least three endoscopic examinations over a period of 3 years, in nine (45.0%) of them, the A/C ratio changed above 15%.

## 4. Discussion

Our study showed that flexible endoscopic adenoid assessment is related to real adenoid size evaluated during its surgical removal; 97.3% sensitivity and 72.7% specificity of flexible endoscopic examination were obtained in the assessment of the size of the pharyngeal tonsils. Inaccuracy of the flexible adenoid assessment during the endoscopic examination may be caused by incomplete relaxation of the soft palate. In our experience, we recommend to the child to blow their nose (“blow the camera”) when we reach the adenoid view. Discomfort with the examination caused the children to want to help us and remove the endoscope by trying to blow their nose. It causes soft palate relaxation and gives us the opportunity to accurately measure the adenoid size when the soft palate is relaxed. This makes the examination more reproducible. A comparison with the lateral X-rays should also be performed at the end of the inspiration, but in that case, it is really difficult to take a single picture at the right moment when we examine a frightened and noncooperating child [18]. Lateral cephalograms reached 61% to 75% sensitivity and 41% to 96% specificity [7,19]. A systematic review performed by Major suggested that lateral X-rays overestimated the size of the adenoid, and should be used for the measurement of the size of the airway rather than the adenoid size [19]. This technique is static and produces a two-dimensional summation picture [14,18,19]. On the other hand, the lateral cephalometric radiograph is a simple, inexpensive, and sufficiently informative diagnostic technique with a low radiation dose [14,18]. A comparative study performed by Mlynarek showed that, in contrast to FNE, lateral X-ray measurements, such as adenoid thickness or A/C ratio, did not correlate with obstructive symptom score [20]. From the other side, Caylakli, in a performed study, found a correlation between the achieved results of lateral X-rays and FNE [21]. Handelman sequentially analyzed 12 children, from 9 months of age to 18 years, using lateral cephalometric radiographs [15]. His study allowed describing the adenoid development in a single patient, and found the greatest obstruction of nasopharyngeal space during pre- and early school children. However, the effectiveness of surgical treatment presented in this work undermines the reliability of the radiological assessment of the tonsil itself. In 5 out of 10 operated children, the authors reported no complete removal of the tonsil in the radiological examination. This is much more than that described by other authors assessing the surgical effects of adenoidectomy (e.g., with endoscopic examination) [22]. Moreover, it has also been proven that evaluation of adenoid size by cephalometric radiography is much less effective than CT, and what was mentioned above overestimated the adenoid size [19,23]. Unfortunately, CT and cone-beam CT have disadvantages, such as high radiation exposure and high cost, which may disqualify them from being performed repeatedly [24]. Analyses of CT examinations in 200 children of different ages divided into five subgroups were carried out by Cohen. He reported involution of adenoid tissue in a group of children between 5.1 and 8 years old. However, this is quite a long period when it comes to observing and deciding on the treatment of the patient [25]. In addition, the reduction of adenoid symptoms and increase of nasal air flow, which are commonly observed at about 7 years of age, may not be related to adenoid involution, but to the expansion of bony nasopharynx confirmed in lateral cephalometry [25,26,27]. Bergland reported an increase of 38% of nasopharyngeal space from 6 years of age to maturity [28]. A study performed by Papaioannou involving the analysis of the size of the pharyngeal tonsil in an MRI study of children of different ages showed that in children who do not snore, the size of the tonsil increases up to 7 to 8 years of age, and then it slowly decreases. In the group of children who snore (more than 1 night per week), the reduction of the tonsil occurred very slowly until 18 years of age [29]. On one hand, this study may indicate that such a slow process of tonsil involution occurs in children presenting to the ENT clinic; on the other hand, the discrepancy in results and curves in both groups—snorers and non-snorers—may derive from the analysis of a too-small group of snoring children: 33 compared with the bigger sample of 149 non-snoring children. It is particularly important in the performed study to analyze large groups of patients, because each patient’s adenoid size was examined only once, and then it was compared with other age patients’ adenoids to analyze the adenoid development from the perspective of years. It is known that in different children, the size of the tonsil is different, which has already been confirmed by Handelman and Pruzansky’s report and numerous subsequent works [30]. Research on ultrasonography of the adenoids is promising, but is still not very common [31]. In contrast to the imaging techniques, invasive diagnostic tools not only provide information about anatomical structures of the nose and nasopharynx, but also visualize the functional state of the nasopharynx. Some of them show the color, mucous coverage, or inflammation characteristics. Invasive methods can cause discomfort and pain for the patient and require his or her collaboration. Otherwise, there is a need for the use of general anesthesia. Video fluoroscopy has good sensitivity—100%—and specificity of 90%, but it produces a 260 microsievert irradiation dose [7,19]. Flexible endoscopy seems to be the least traumatic of the invasive techniques and, as we have shown, in experienced hands, may be performed without anesthesia and gives plenty of information about adenoid state [32,33]. In the era of commonly performed COVID-19 tests, this examination is less painful than a nasal swab, in the opinion of patients.

Sensitivity and specificity of the adenoid size assessment during the surgery may also be burdened with an incomplete resection of the adenoid. Lesinskas reported 31.3% adenoid regrowth in children younger than 5 years old [34]. The reasons for the regrowth may be incomplete resection or persistent infections of the upper respiratory tract in postsurgical children’s lives, asthma, gastroesophageal reflux (GERD), and allergic rhinitis. [34,35]. It is also dependent on the surgeon’s experience and surgical technique [22,34]. Yildirim shows that blind curettage adenoidectomy may have left in 18% large residual adenoid. For precise resection of the adenoid, we controlled the nasopharynx with a mirror [22]. In our study, one patient (fewer than 1%) needed revision surgery because of adenoid regrowth (A/C ratio 70%) and concomitance hypoacusis caused by middle-ear effusion. This might be consistent with Dearking’s observation that children with ear-related indications and obstructive adenoid symptoms were significantly likelier to require revision adenoidectomy [36]. The overall rate of revision adenoidectomy is estimated from 1.6% to 2.5% [35,37]. In the performed study, we obtained similar effectiveness of treatments compared to other authors, which validates the proper surgical technique and precise adenoid removal.

This study verified the gold standard in adenoid examination by comparing the endoscopic adenoid assessment with the real size of the removed adenoid. Moreover, it shows that an adenoid measured in endoscopic examination as a 75% A/C ratio is assessed intraoperatively as a large adenoid.

Our research showed that in performing the first endoscopic examination and reporting an A/C ratio below 60%, we have about a 26% probability that within 1 year and 21% in a 2-year period, the tonsil will reach 75% A/C ratio and it will be assessed during the surgical procedure as large. In such children, an examination to assess the size of the tonsil should not be performed more frequently than 3 years if adenoid symptoms do not increase. In these children, conservative treatment of adenoid hypertrophy symptoms should be applied. However, in the group of children with a baseline tonsil of 60–70% in A/C ratio, this examination should be performed more often, every 2 years, if the adenoid hypertrophy symptoms do not intensify. In case of an increase in those symptoms, FNE should be performed every year. Children whose tonsil size exceeds 70% in A/C ratio should also be controlled every year. As mentioned above, there are many statements about the time of involution of the adenoid in children in the specialized literature, but there are few studies that confirm these theses [13,14,15,31]. Ishida performed sequential lateral cephalometric radiography in 90 children from 6 to 19 years old over a 10-year period of time. This performed study shows slow multiannual tendency of adenoid involution, but there was no decrease in adenoid size among groups of children in neighboring groups (for example: lower primary school, 8 years old, vs. upper primary school children, 10 years old) [14]. As mentioned earlier, lateral X-rays overestimated the size of the adenoid and should be used for the measurement of the size of the airway rather than the adenoid [19]. However, this study proves the tendency toward very slow adenoid involution. Yamada performed a similar retrospective study in Japan. He analyzed, in a 5-year period of time, a sample involving 99 individuals of the ages of 8–12 years. He stated that the adenoid-to-nasopharynx index decreased significantly through elementary school (in Japan, ages 6 to 12 years) [13]. However, the most interesting are the studies concerning children aged 3 to 8 years, because at these ages, adenoid symptoms seem to be the most burdensome for children. Wang analyzed the adenoid change in an ultrasound measurement of adenoid thickness in children aged 3 to 12 years and showed that the mean value of ultrasound measurements of adenoids in children aged 6 years was significantly greater than that of children of other ages [31]. We performed the first analysis of adenoid size change in periods of time in children 2.5 to 8 years old with the use of FNE. Based on our study, we propose a schema for the frequency of FNE for adenoid size assessment and adenoid hypertrophy treatment (Figure 3).

A limitation of this study was the selection of a group of patients undergoing long-term observation. After the diagnosis, parents were informed about the possibility of conservative or surgical treatment of adenoid hypertrophy in their children. Conservatively-treated children received a 12-week course treatment with mometasone furoate nasal spray and saline irrigation, which is a standard pharmacological treatment for adenoid hypertrophy symptoms, hoping for improvement [38]. We should add that our newest study did not reveal any change in adenoid size 3 to 6 months after finishing a 12-week course of intranasal steroid treatment, but in other studies, we showed seasonal variability of symptoms [33,39]. The need to wait for the adenoidectomy procedure in our country and, in some cases, the temporary (seasonal) relief of symptoms made such an observation possible until the procedure of adenoidectomy. In some cases, parents did not decide to undergo surgery because of symptom relief or concerns about surgical complications. This meant that, with the follow-up period, the number of patients decreased. The influence on the size of the sample is the fact that those groups were examined and also operated on by one children’s ENT specialist (A.Z.) in the same ENT outpatient clinic and hospital. On the other hand, it affects the repeatability of the tests performed by the same doctor using the same flexible endoscopic system and surgical technique.

## 5. Conclusions

FNE examination is highly effective in assessing the size of the pharyngeal tonsil—a high compliance of endoscopic examination with intraoperative assessment of the size of the tonsil has been demonstrated. In contrast to imaging tests, such as X-ray or ultrasound, this examination determines not only the size of the tonsil, but also its mucous coverage, edema, and inflammation status. Sensitivity and specificity of the flexible nasopharyngoscopy were calculated as 97.3% and 72.7%, respectively. We confirmed that a 75% A/C ratio or more is equivalent with an intraoperative large adenoid. Failure to remove a large adenoid during surgery in children who had a high A/C ratio before surgery may be a signal to the surgeon that he or she did not cut the whole tonsil, especially if he or she performed a blind curettage adenoidectomy. We proposed a schema for frequencies of FNE examinations and treatment depending on A/C ratio and worsening of ailments.

## Figures and Tables

**Figure 1 diagnostics-12-01734-f001:**
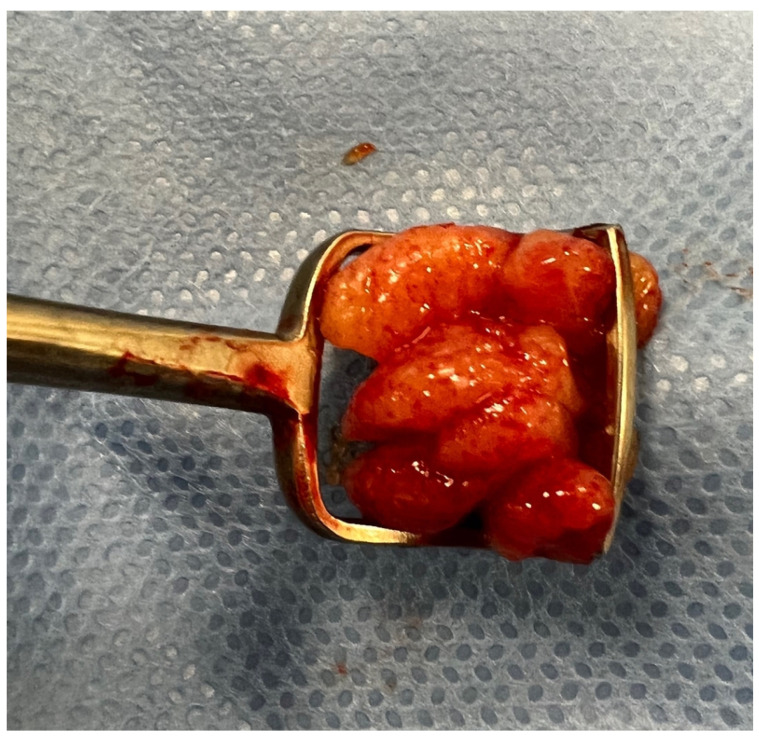
Intraoperative adenoid size measurement.

**Figure 2 diagnostics-12-01734-f002:**
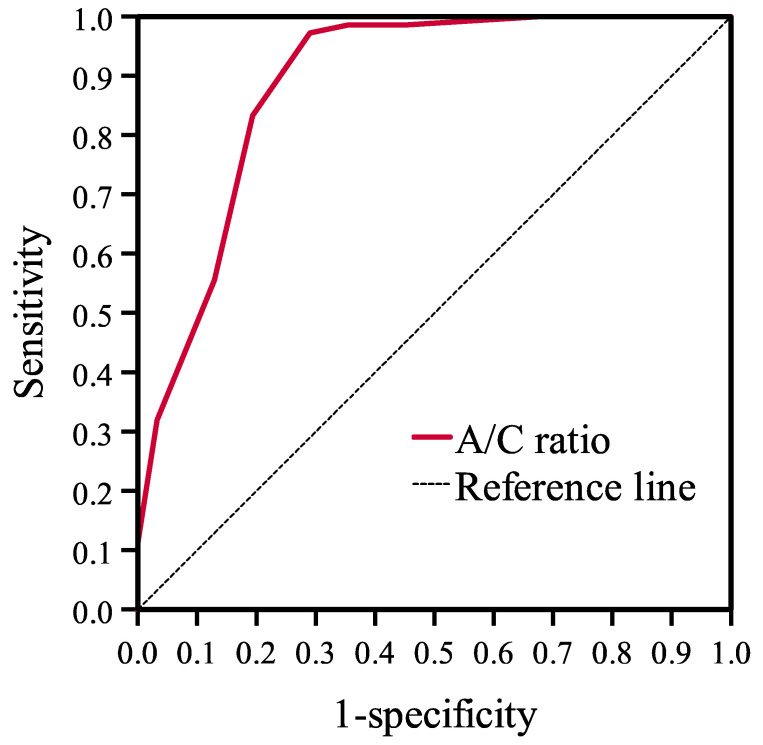
Diagnostic accuracy of A/C ratio for diagnosis of large operative adenoid size.

**Figure 3 diagnostics-12-01734-f003:**
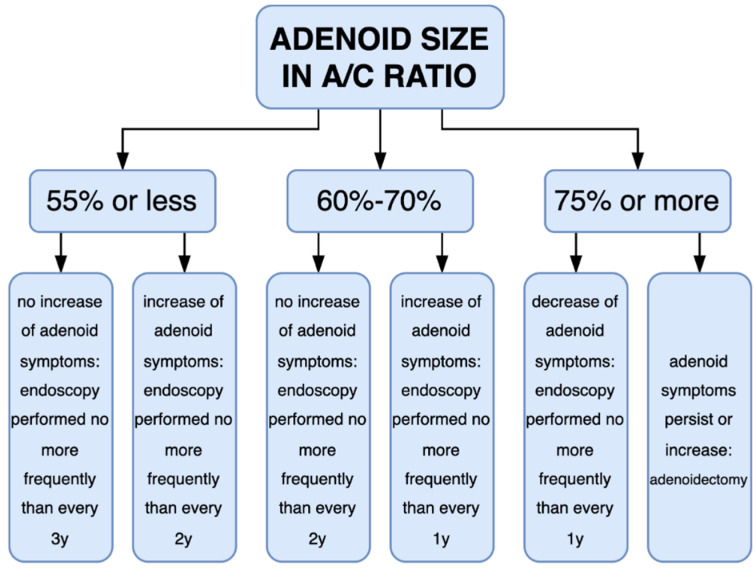
Schema of frequency of adenoid examinations and treatment.

**Table 1 diagnostics-12-01734-t001:** Patients’ characteristics in the whole first group who underwent adenoidectomy and according to gender.

Characteristics		All Patients	Female	Male	*p* Value
*n*		108	43	65	
Gender	female	43 (39.8%)	43 (39.8%)		-
	male	65 (60.2%)		65 (60.2%)	
Age (years)	mean ± *SD*	5.2 ± 1.8	5.4 ± 1.9	5.0 ± 1.7	0.267
	median (Q25–Q75)	5.0 (4.0–6.0)	5.0 (4.0–6.0)	4.5 (4.0–5.5)	
Adenoid size (A/C ratio *, %)	mean ± *SD*	77.5 ± 13.2	78.8 ± 13.4	76.6 ± 13.2	0.396
	median (Q25–Q75)	80.0 (75.0–85.0)	80.0 (75.0–90.0)	80.0 (70.0–85.0)	
	40	1 (0.9%)	1 (2.3%)	0 (0.0%)	0.505
	50	10 (9.3%)	2 (4.7%)	8 (12.3%)	
	60	9 (8.3%)	4 (9.3%)	5 (7.7%)	
	65	3 (2.8%)	1 (2.3%)	2 (3.1%)	
	70	3 (2.8%)	1 (2.3%)	2 (3.1%)	
	75	13 (12.0%)	7 (16.3%)	6 (9.2%)	
	80	23 (21.3%)	7 (16.3%)	16 (24.6%)	
	85	22 (20.4%)	7 (16.3%)	15 (23.1%)	
	90	16 (14.8%)	8 (18.6%)	8 (12.3%)	
	95	8 (7.4%)	5 (11.6%)	3 (4.6%)	
Adenoid size (A/C ratio on Bolesławska scale, %)	35–65 (B II)	23 (21.3%)	8 (18.6%)	15 (23.1%)	0.587
	>65 (B III)	85 (78.7%)	35 (81.4%)	50 (76.9%)	
Operative adenoid size, 3 categories	small	21 (19.4%)	8 (18.6%)	13 (20.0%)	0.857
	medium	12 (11.1%)	6 (14.0%)	6 (9.2%)	
	large	75 (69.4%)	29 (67.4%)	46 (70.8%)	
Operative adenoid size, 2 categories	not large	33 (30.6%)	14 (32.6%)	19 (29.2%)	0.713
	large	75 (69.4%)	29 (67.4%)	46 (70.8%)	
Consistency between A/C ratio in Bolesławska scale and operative adenoid size **	true	96 (88.9%)	37 (86.0%)	59 (90.8%)	0.536
	false	12 (11.1%)	6 (14.0%)	6 (9.2%)	

* A/C ratio—adenoid-to-choana ratio. ** Consistency when second degree in the Bolesławska scale (B II) = not large operative adenoid size (small and medium) or B III = large operative adenoid size.

**Table 2 diagnostics-12-01734-t002:** Association between macroscopic adenoid size and demographic variables and A/C ratio.

Characteristics	Operative Adenoid Size, 3 Categories			*p* Value ^1^	*p* Value ^2^
	Small	Medium	Large		
*n*	21	12	75		
Gender					
female	8 (38.1%)	6 (50.0%)	29 (38.7%)	0.857	0.713
male	13 (61.9%)	6 (50.0%)	46 (61.3%)		
Age (years)					
mean ± *SD*	5.0 ± 1.5	4.7 ± 1.3	5.3 ± 1.9	0.559	0.336
median (Q25–Q75)	4.5 (4.0–6.5)	4.3 (4.0–5.8)	5.0 (4.0–6.0)		
Adenoid size (A/C ratio, %)					
mean ± *SD*	59.5 ± 13.6	69.6 ± 10.1	83.8 ± 6.8	<0.001 ^3^	<0.001
median (Q25–Q75)	50.0 (50.0–65.0)	67.5 (60.0–77.5)	85.0 (80.0–90.0)		
Adenoid size (A/C ratio on Bolesławska scale, %)					
35–65 (B II)	16 (76.2%)	6 (50.0%)	1 (1.3%)	<0.001	<0.001
>65 (B III)	5 (23.8%)	6 (50.0%)	74 (98.7%)		
Consistency between A/C ratio on Bolesławska scale and operative adenoid size					
true	16 (76.2%)	6 (50.0%)	74 (98.7%)	<0.001	<0.001
false	5 (23.8%)	6 (50.0%)	1 (1.3%)		

^1^ *p* value for comparison of small vs. medium vs. large adenoid size; ^2^ *p* value for comparison of not large (small + medium) vs. large adenoid size. ^3^ Bonferroni post hoc tests: small vs. medium, *p* = 0.007; medium vs. large, *p* < 0.001; small vs. large, *p* < 0.001.

**Table 3 diagnostics-12-01734-t003:** Diagnostic value of A/C ratio for diagnosis of large operative adenoid size.

	A/C Ratio	
**AUC**	0.894	
**95% CI**	0.818–0.969	
***p* value**	<0.001	
**Cutoff value (%)**	75.0	80.0
**Sensitivity**	97.3%	84.0%
**Specificity**	72.7%	81.8%
**PPV**	89.0%	91.3%
**NPV**	92.3%	69.2%

AUC—area under the curve. 95% CI—95% confidence interval. PPV—positive predictive value. NPV—negative predictive value.

**Table 4 diagnostics-12-01734-t004:** Characteristics of the group of patients undergoing long-term observation.

Characteristics		All Patients
*n*		81
Gender	female	43 (53.1%)
	male	38 (46.9%)
Age at the first visit (years)	mean ± *SD*	3.9 ± 1.2
	median (Q25–Q75)	3.5 (3.0–4.5)
Number of measurements	3	47 (58.0%)
	4	16 (19.8%)
	5	15 (18.5%)
	6	2 (2.5%)
	7	1 (1.2%)
Period of observation (years)	mean ± *SD*	2.4 ± 1.2
	median (Q25–Q75)	2.0 (1.5–3.0)
	1–3	63 (77.8%)
	3.5–5.5	18 (22.2%)
First visit	mean ± *SD*	65.4 ± 13.2
	median (Q25–Q75)	65.0 (55.0–75.0)
Bolesławska scale	<35	1 (1.2%)
	35–65	46 (56.8%)
	>65	34 (42.0%)
Last visit	mean ± *SD*	61.2 ± 16.1
	median (Q25–Q75)	60.0 (50.0–75.0)
Bolesławska scale	<35	4 (4.9%)
	35–65	46 (56.8%)
	>65	31 (38.3%)
Change; first vs. last visit	decrease	21 (25.9%)
	no change	17 (21.0%)
	increase	43 (53.1%)
	≤10	54 (66.7%)
	>10	27 (33.3%)
	≤15	61 (75.3%)
	>15	20 (24.7%)
Max	mean ± *SD*	71.0 ± 12.3
	median (Q25–Q75)	70.0 (60.0–80.0)
Min	mean ± *SD*	56.3 ± 14.6
	median (Q25–Q75)	55.0 (50.0–70.0)
Range	mean ± *SD*	14.7 ± 11.2
	median (Q25–Q75)	10.0 (5.0–20.0)
	≤10	42 (51.9%)
	>10	39 (48.1%)
	≤15	56 (69.1%)
	>15	25 (30.9%)
Semi-annual continuous measurements, 1 year	*n*	38 (46.9%)
Range	≤10	28 (73.7%)
	>10	10 (26.3%)
	≤15	35 (92.1%)
	>15	3 (7.9%)
Annual continuous measurements (at least one measurement per year)	*n*	57 (70.4%)
Range	≤10	34 (59.6%)
	>10	23 (40.4%)
	≤15	42 (73.7%)
	>15	15 (26.3%)
Two-year period	*n*	37 (45.7%)
Range	≤10	21 (56.8%)
	>10	16 (43.2%)
	≤15	29 (78.4%)
	>15	8 (21.6%)
Three-year period	*n*	20 (24.7%)
Range	≤10	7 (35.0%)
	>10	13 (65.0%)
	≤15	11 (55.0%)
	>15	9 (45.0%)

## Data Availability

Additional data supporting reported results may be available on request.

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
