# Peer review of "Effectiveness of Evaluation of Adenoid Hypertrophy in Children by Flexible Nasopharyngoscopy Examination (FNE), Proposed Schema of Frequency of Examination: Cohort Study"

_diagnostics, 2022, doi:10.3390/diagnostics12071734_

Round 1

Reviewer 1 Report

At the outset, I would like to congratulate the authors for submitting their work. The subject matter is important. However, I have a few comments that need to be addressed by the authors:

-Introduction: What did you hypothesize before conducting this research? Please add your hypothesis in 2-3 lines at the end of the Introduction section.

-Methods: What is the study design? Were both the time periods retrospective? Please clearly mention the study design and give a clear description of the time periods.

-Results: Well written.

Was the time duration between endoscopic examination and surgery constant? could this duration affect the measurements? 

-Discussion: It is very lengthy. Please remove unnecessary paragraphs that are not specific to this research subject (I can see there are many paragraphs on general information).

Please add a paragraph on the limitations of this manuscript.

Author Response

Dear Reviewers,

I’d like to thank you for accurate analysis of my text.

Alexander Zwierz

At the outset, I would like to congratulate the authors for submitting their work. The subject matter is important. However, I have a few comments that need to be addressed by the authors:

-Introduction: What did you hypothesize before conducting this research? Please add your hypothesis in 2-3 lines at the end of the Introduction section. We add the hypothesis -line 71-73

-Methods: What is the study design? Were both the time periods retrospective? Please clearly mention the study design and give a clear description of the time periods. Both are the retrospective analysis -line 77

-Results: Well written. 

Was the time duration between endoscopic examination and surgery constant? could this duration affect the measurements? Yes, the time between endoscopy and surgery was up to 1 month.

-Discussion: It is very lengthy. Please remove unnecessary paragraphs that are not specific to this research subject (I can see there are many paragraphs on general information). It is difficult for the author to remove his own sentences, some paragraphs because it will disturb the concept of the discussion. Please suggest the paragraphs to remove.

Please add a paragraph on the limitations of this manuscript.

Paragraphs has been added line 331

Reviewer 2 Report

The topic is very interesting, but the quality of the manuscript is poor. The manuscript is poorly written and it is very hard to understand, which is not clear to me if it is due to the poor writting or due to the fact that the authors did no state clearly the methods and research design. It is clear that endoscopic assessment is extremely useful for adenoids, but the authors should definitely refer the manuscript to well trained physicians who has experience in publishing for reorganizing their work and correct the scientific aspects, and afterwards to a professional proofreading service or a native English speaker.

Author Response

Dear Reviewer,

The topic is very interesting, but the quality of the manuscript is poor. The manuscript is poorly written and it is very hard to understand, which is not clear to me if it is due to the poor writting or due to the fact that the authors did no state clearly the methods and research design. It is clear that endoscopic assessment is extremely useful for adenoids, but the authors should definitely refer the manuscript to well-trained physicians who has experience in publishing for reorganizing their work and correct the scientific aspects, and afterwards to a professional proofreading service or a native English speaker.

It is difficult to dispute about style of the manuscript, which in other reviewer opinion is was “easy to read and follow” and has been checked by the scribendi language editors. It was checked once again. We get the editing certificate. Please indicate what you don’t understand in merits.

Reviewer 3 Report

The Authors will carefully reread the manuscript: 1) there are various typing errors; 2) at least one acronym to be fully detailed (OME, line 50); 3) some references to be corrected. For example: a) Handelman (reference 15): 1974 in the text, 1976 in references; b) Daerking or Dearking (reference 37)? c) Lesinskas, reference 34, but 33 in the text.

Author Response

Dear Reviewer,

I’d like to thank you for accurate analysis of my text.

Alexander Zwierz

The Authors will carefully reread the manuscript: 1) there are various typing errors; 2) at least one acronym to be fully detailed (OME, line 50); 3) some references to be corrected. For example: a) Handelman (reference 15): 1974 in the text, 1976 in references; b) Daerking or Dearking (reference 37)? c) Lesinskas, reference 34, but 33 in the text. We correct the errors. The text was reedit by profesional editing service- once again.

Round 2

Reviewer 1 Report

In the revised manuscript, all my comments have been addressed. The overall scientific quality of the manuscript has improved significantly. I would like to congratulate the authors for their work.

Author Response

Thank you very much fr your work to improve my manuscript.

Reviewer 2 Report

The authors considerably improved the quality of their manuscript. Nevertheless, several minor mistakes are still present (e.g. 'Conservativly'). Moreover, the authors should define more precisely the limitations of this study.

Author Response

Thank you for your comments, I developed the section about limitations of the study, have corrected the mistakes. 

Alexander